# Monitoring Metastatic Colorectal Cancer Progression According to Reactive Oxygen Metabolite Derivative Levels

**DOI:** 10.3390/cancers15235517

**Published:** 2023-11-22

**Authors:** Katsuji Sawai, Takanori Goi, Youhei Kimura, Kenji Koneri

**Affiliations:** First Department of Surgery, University of Fukui, Fukui 910-1193, Japan; tgoi@u-fukui.ac.jp (T.G.); y-kimura@kimura-hospital.jp (Y.K.); koneri@u-fukui.ac.jp (K.K.)

**Keywords:** colorectal adenocarcinoma, tumor marker, treatment response, chemotherapy, oxidative stress

## Abstract

**Simple Summary:**

Oxidative stress has been implicated in the development, proliferation, and metastasis of colorectal cancer. In this study, we investigated whether the rate of change in reactive oxygen metabolite derivatives (d-ROM)—serum markers of oxidative stress—could predict treatment response in metastatic colorectal cancer. We enrolled 53 patients with metastatic colorectal cancer who were treated with 3 months of chemotherapy. We measured d-ROM levels before and after chemotherapy and examined the change in d-ROM levels for each anticancer treatment. Factors influencing the d-ROM ratio (post-treatment: pre-treatment levels) were examined using linear regression analysis. d-ROM levels decreased in patients showing a partial response and increased in those showing disease progression. An increasing d-ROM ratio was associated with disease progression. Our study indicates that d-ROM levels are useful markers of tumor progression and that the d-ROM ratio is useful for predicting treatment response in patients with metastatic colorectal cancer.

**Abstract:**

Oxidative stress has been implicated in the development, proliferation, and metastasis of colorectal cancer, but few studies have considered how oxidative stress changes in relation to treatment response. In this study, we investigated whether the rate of change in reactive oxygen metabolite derivatives (d-ROM)—serum markers of oxidative stress—could predict treatment response in metastatic colorectal cancer. We enrolled 53 patients with metastatic colorectal cancer who were treated with 3 months of chemotherapy. We measured d-ROM levels and performed computed tomography before and after chemotherapy, and we examined the change in d-ROM levels for each anticancer treatment. Factors influencing the d-ROM ratio (post-treatment: pre-treatment levels) were examined using linear regression analysis. d-ROM levels decreased in patients showing a partial response (*p* < 0.001) and increased in those showing disease progression (*p* = 0.042). An increasing d-ROM ratio was associated with disease progression (regression coefficient: 0.416, 95% confidence interval: 0.279–0.555, *p* < 0.001). Our study indicates that d-ROM levels are useful markers of tumor progression and that the d-ROM ratio is useful for predicting treatment response in patients with metastatic colorectal cancer.

## 1. Introduction

In Japan, the number of colorectal cancer cases increases annually, and the number of deaths from colorectal cancer currently exceeds 50,000, making it the second leading cause of cancer-related deaths [1]. Moreover, approximately 30% of patients with colorectal cancer die because of metastasis [2].

The prognosis of metastatic colorectal cancer has markedly improved with the recent introduction of new anticancer therapies, such as molecular-targeted agents [3]. However, effective long-term disease management requires the continued evaluation of treatment efficacy. Treatment efficacy could be determined using computed tomography (CT) imaging with the Response Evaluation in Solid Tumors (RECIST) criteria (recommended by the World Health Organization [WHO]) or by assessing changes in carcinoembryonic antigen (CEA) levels (recommended by the National Comprehensive Cancer Network [NCCN] guidelines) [4,5,6,7,8,9]. However, imaging has its limitations: it is not particularly cost-effective and is ineffective in the absence of measurable or disseminated lesions. CEA is also only an effective marker of treatment efficacy when its levels are abnormal before treatment [10,11].

Oxidative stress plays a critical role in various biological processes, particularly in cell signaling. Under normal conditions, it is balanced by cellular antioxidant defenses. However, an imbalance leading to oxidative stress overload can cause significant harm, including extensive DNA damage, and the oxidation of key lipids and proteins. These changes disrupt intracellular signaling, contributing to abnormal cell functioning. Such disruption is a major factor in promoting carcinogenesis, accelerating cell mutation, and impairing cell function, thereby facilitating the development and progression of various cancers [12,13]. Increased oxidative stress is often triggered by the activation of nicotinamide adenine dinucleotide phosphate (NADPH) oxidase in neutrophils’ plasma membranes and the production of lipid peroxidation products in cancer tissues [14,15,16]. This stress influences the activities of matrix metalloproteinases, adhesion molecules, and the epidermal growth factor and its receptor, which are instrumental in cancer progression and metastasis [17,18,19]. Specifically, oxidative stress has been linked to the development of colorectal cancer. The transformation of colorectal adenomas into colorectal cancer involves the activation of oncogenes, inactivation of tumor suppressor genes, and DNA methylation. These DNA mutations, primarily induced by oxidative stress, underscore its significant impact in the pathogenesis of cancer [20]. Quantitative methods for measuring stable oxidative stress in blood, such as via reactive oxygen metabolite derivative (d-ROM) measurement, have recently been developed [21]. Blood d-ROM levels tend to increase with tumor size and progression, and patients with high blood d-ROM levels have been associated with poor disease prognosis [16,18,22]. However, the effect of anticancer treatment on d-ROM levels in colorectal cancer is unclear.

In this study, we tested the hypothesis that d-ROM levels decrease in colorectal cancer patients with unresectable lesions when treatment is effective and increase when treatment is ineffective. Consequently, we verified the use of oxidative stress as a marker of treatment efficacy in patients with unresectable colorectal cancer.

## 2. Materials and Methods

### 2.1. Study Population

We analyzed data for 53 patients with unresectable metastatic colorectal cancer and unresectable recurrent colorectal cancer who had undergone chemotherapy or management therapy at the department of the University of Fukui between January 2020 and December 2021. Patients with synchronous or heterochronic cancer, coexisting infectious disease, steroid therapy, and/or lack of follow-up data were excluded. Written informed consent was obtained from participants prior to the study.

### 2.2. Treatment and Response Evaluation by CT

Chemotherapy was administered according to the colorectal cancer treatment guidelines and included two-drug combination therapy, three-drug combination therapy, oral anticancer drugs, and molecular-targeted drugs [1]. Four patients were not treated with anticancer drugs. To evaluate treatment efficacy (for both patients treated with and without anticancer drugs), CT scans were performed every 3 months, and images before and 3 months after the start of treatment were evaluated according to the RECIST criteria.

### 2.3. Measurement of d-ROM Levels

Using blood samples collected during routine examinations, d-ROM levels were measured before and 3 months after the start of treatment.

Serum d-ROM levels were determined using the FREE radical analyzer system by Wismerll Co., Ltd., Tokyo, Japan, which includes a spectro-photometric device. This process involves using both a reader and specific measurement kits (d-ROMs test, Wismerll Co., Ltd.), all of which are calibrated for optimal performance with the FREE Carpe Diem System, as per the guidelines provided by the manufacturer. Initially, a 20 µL sample of serum is combined with 1 mL of a specially formulated buffered solution in a cuvette. Subsequently, 20 µL of a chromogenic substrate is introduced into this mixture. Upon thorough mixing, the cuvette is then inserted into the thermostatic block of the analyzer. The sample is incubated here for 5 min at a controlled temperature of 37 °C. Post-incubation, the absorbance is measured at a wavelength of 505 nm. The results from this procedure are presented in U.CARR units, with one unit equating to 0.8 mg/L of hydrogen peroxide. Typically, the normal range for these measurements is between 250 to 300 U.CARR. Any d-ROM level that equals or exceeds 300 U.CARR is indicative of serum oxidative stress, a condition attributed to the excessive production of free radicals in the body [23,24,25,26].

### 2.4. Assessment of d-ROM Levels

We examined the pre- and post-treatment changes in d-ROM levels in the partial response (PR), stable disease (SD), and progressive disease (PD) groups using the RECIST criteria. We examined the d-ROM ratio (i.e., the ratio of d-ROM levels before and after treatment) and determined the influencing factors using linear regression analysis. We set a cutoff value for the d-ROM ratio to predict disease progression and assessed the positive predictive value, negative predictive value, and diagnostic accuracy.

### 2.5. Statistical Analysis

Changes in d-ROM levels and CT image RECIST scores before and after treatment were examined using a Wilcoxon signed rank test. We performed a multiple linear regression analysis to identify factors affecting the d-ROM ratio, including age, sex, treated or not, number of metastatic organs, use of molecular-targeted drug, use of oxaliplatin-based regimens, and the RECIST criteria. We assessed the normality of residuals obtained in the final model using a normal probability plot. Receiver operating characteristic (ROC) curve analysis was performed to estimate the optimal sensitivity and specificity of the d-ROM ratio in predicting PD based on the RECIST score. All statistical analyses were performed using IBM SPSS software version 21.0 (IBM Japan, Ltd., Tokyo, Japan). Differences were considered significant at *p* < 0.05.

## 3. Results

### 3.1. Patient Characteristics

The patient group (n = 53, 30 males and 23 females) had a median age of 69 years (range: 38–93 years). Regarding treatments, 44 patients (83%) had a resection of the primary tumor, 47 (88.7%) had concurrent metastases, and 37 (69.8%) had no prior anticancer therapy. Regarding the metastatic organ count, 28 patients (52.8%) had one metastatic organ, and 25 (47.2%) had more than one (Table 1).

### 3.2. Anticancer Drug Treatment and Effects

A doublet oxaliplatin-based anticancer regimen was applied for 33 patients, doublet irinotecan-based regimen for 5 patients, triplet regimen for 4 patients, 5-fluorouracil-based oral anticancer drugs for 5 patients, and trifluridine/tipiracil hydrochloride for 2 patients. A total of 44 patients received molecular-targeted agents (Table 2). According to the RECIST criteria, 26 patients (49%) had a PR, 10 patients (18.9%) had an SD, and 17 patients (32.1%) had a PD (Table 3). The median d-ROM level before treatment was 424 U.CARR (range: 283–636), and the median d-ROM level after 3 months was 375 U.CARR (range: 168–664) (Table 4).

### 3.3. Relationship between Treatment Response by the RECIST Criteria and Changes in d-ROM Levels

In PR patients, d-ROM levels significantly decreased after treatment compared to those before treatment (*p* < 0.001). d-ROM levels in PD patients increased (*p* = 0.042), while those in SD patients were unchanged (*p* = 0.139). In non-PD patients (PR and SD patients combined), d-ROM levels significantly decreased after treatment compared to those before (*p* < 0.001) (Figure 1). Non-PD patients (median: 437 U.CARR) had higher pre-treatment d-ROM levels than PD patients (median: 381 U.CARR; *p* = 0.029). Post-treatment d-ROM levels did not differ between non-PD (median: 362 U.CARR) and PD (median: 423) patients (*p* = 0.051).

### 3.4. Accuracy of the d-ROM Ratio in Predicting Disease Progression

Linear regression analysis showed that the d-ROM ratio varied significantly with RECIST-rated PD (regression coefficient: 0.416, 95% confidence interval [CI]: 0.277–0.555, *p* < 0.001), oxaliplatin use (regression coefficient: 0.209, 95% CI: 0.063–0.354, *p* = 0.006), and female sex (regression coefficient: −0.111, 95% CI: −0.205 to 0.017, *p* = 0.021) (Table 5). Age, being treated or not, the number of metastatic organs, and use of molecular-targeted drugs did not affect the d-ROM ratio. The cutoff d-ROM ratio for predicting a PD was 1.006, corresponding to an area under the curve of 0.874 with a 95% CI of 0.0760–0.988 (*p* < 0.001) (Figure 2). Therefore, the cutoff value was set at 1.0. There were 36 patients with a d-ROM ratio ≤ 1, of which 33 were non-PD patients; 17 had a d-ROM ratio > 1, of which 14 were PD patients (Table 4). When predicting a PD with a d-ROM ratio > 1, the sensitivity was 82.4%, specificity was 91.7%, positive predictive value was 82.4%, negative predictive value was 91.7%, and diagnostic accuracy was 88.7% (Table 6). 

## 4. Discussion

In this study, we examined the relationship between changes in d-ROM levels and chemotherapeutic treatment outcomes. Two important findings were obtained in this study. First, in patients with metastatic colorectal cancer, blood d-ROM levels decreased when treatment was effective and increased with tumor progression. Second, the d-ROM ratio was highly sensitive and specific in predicting disease progression and is therefore considered to be an effective marker for monitoring treatment outcomes in this patient population.

The treatment of non-resectable colorectal cancer is generally aimed at suppressing tumor growth with anticancer drugs, alleviating tumor-related symptoms, and, if possible, reducing tumor size to allow for resection. Therefore, new tumor markers need to be identified to address the shortcomings of traditional methods. Colon tissues are constantly exposed to intrinsic and extrinsic oxidative stress, which can promote carcinogenesis through protein denaturation, DNA damage, and lipid oxidation, promoting tumor growth, invasion, and metastasis through various metabolites [27,28,29,30]. Blood oxidative stress levels are higher in patients with colorectal cancer than in healthy individuals [15,31,32], and they show a strong association with tumor size, progression, and prognosis [16,18,22,32]. Oxidative stress levels are also higher in colon cancer tissues than in the surrounding tissues [15,32,33]. Although a close relationship between oxidative stress and colorectal cancer has been observed, only few studies have assessed changes in oxidative stress to determine the efficacy of treatment for colorectal cancer. The redox state in colorectal cancer is affected by inflammatory infiltration and the environmental characteristics of the tumor. Inflammatory reactions in cancer tissues activate NADPH oxidase in neutrophils, which amplifies oxidative stress. When the redox balance is disturbed in normal cells, antioxidant defense and repair are activated to ameliorate oxidative stress, but in cancer cells, antioxidant defense and repair are not fully activated, resulting in increased oxidative stress. Oxidative stress promotes the expression of matrix metalloproteinases, adhesion molecules, and epidermal growth factor and its receptor, which in turn promotes cancer progression and metastasis. Oxidative stress can promote oxidative damage to DNA, proteins, and lipids, resulting in the production of lipid peroxidation products, such as malondialdehyde (MDA) and oxidized low-density lipoprotein (ox-LDL). MDA severely damages DNA and inhibits its repair through direct interaction with DNA repair-related proteins [34]. ox-LDL is also a potent mitogen, promoting the expression of proteins that activate the cell cycle and releasing cytokines and growth factors, which in turn accelerate cancer progression [34,35,36,37,38]. Cancer cell proliferation requires increased levels of adenosine triphosphate (ATP), which allows cancer cells to absorb glucose at a faster rate and promotes glycolysis. However, inadequate oxygen supply leads to anaerobic metabolism, resulting in the formation of advanced glycation end products (AGEs) [39,40]. Increased production of AGEs can lead to tumor initiation, growth, and invasion through increased cell proliferation and cell migration [41,42]. Furthermore, it is believed that oxidative stress and cancer cell proliferation interact to exacerbate oxidative stress within the tumor.

Salehi et al. measured changes in the levels of the oxidative stress markers MDA, ox-LOL, and AGEs before and after tumor resection in 60 patients with stage I and II colorectal cancer and confirmed a decrease in the levels after resection. They also reported an increase in the levels of antioxidant factors, which was likely related to the resection of tumor tissues with high oxidative stress levels [15]. Acevedo-León et al. measured oxidative stress levels based on the disulfide-oxidized forms of the tripeptide L-γ-glutamyl-cysteinyl-glycine (GSSG) in 79 patients with stage I to IV colorectal cancer and reported a decrease after resection compared to those before [32]. Hristozov et al. measured MDA levels in red blood cells and found a marked decrease after surgery compared to those before [43]. In the present study, we examined changes in tumor response and oxidative stress related to anticancer drug treatment rather than tumor resection. d-ROM levels decreased after treatment in PR patients and increased in PD patients. We confirmed that a PD was strongly associated with an increased d-ROM ratio. Our findings are consistent with the decreased and increased oxidative stress levels following tumor resection and disease progression, respectively, observed in previous studies. However, the effect of anticancer drugs on oxidative stress levels remains inconclusive. Kadam and Abhang reported an increase in oxidative stress with anticancer drug administration in patients with breast cancer, while El-Hefny et al. found no difference in oxidative stress before and after chemotherapy [44,45,46]. Yokoyama et al. investigated the induction of intracellular oxidative stress by anticancer drugs using DLD-1, a colon cancer cell line, and reported that fluorouracil and oxaliplatin could not induce oxidative stress [47]. In the present study, we tested the effects of various anticancer drugs and used a multivariate analysis approach to address the bias associated with the different effects of each drug.

When using oxidative stress levels as tumor markers, a high sensitivity and specificity are needed. Kim et al. reported that the CEA level, currently recommended as a colorectal cancer marker, is not an effective indicator of therapeutic response unless the pre-treatment CEA level is above 5 ng/mL. Wang et al. reported a sensitivity of 81% and specificity of 88% for a 50% or greater increase in the CEA level as an effective predictor of disease progression [10,48]. Huang et al. defined disease progression as an increase in the CEA level above the pre-treatment value and reported a sensitivity of 67.7% and specificity of 78.4% in an analysis of 330 cases [49].

In this study, the d-ROM ratio cutoff value for PD prediction was determined to be 1, based on an ROC curve analysis. A d-ROM ratio > 1 meant that the sensitivity and specificity of PD prediction were 82.4% and 91.7%, respectively, which were superior compared to the sensitivity and specificity of CEA in various past studies. The d-ROM ratio is an easy-to-calculate marker of tumor progression that could aid in determining the efficacy of chemotherapy.

We acknowledge some limitations to our study. First, the study of the d-ROM ratio was limited to stage IV colorectal cancer, and it is unclear whether its performance would remain constant at all stages and among cancer types. Second, d-ROM levels may be affected by inflammatory reactions, such as co-infections. In addition, as we compared d-ROM levels in the same patient, the influence of lifestyle and other factors is considered to be minimal, but the influence of diet remains unclear.

## 5. Conclusions

A significant correlation was demonstrated between the rate of change in d-ROM levels and treatment response in patients with metastatic colorectal cancer. This insight opens avenues for further research into the role of oxidative stress in cancer treatment and the potential utility of d-ROM levels as a biomarker in clinical practice.

## Figures and Tables

**Figure 1 cancers-15-05517-f001:**
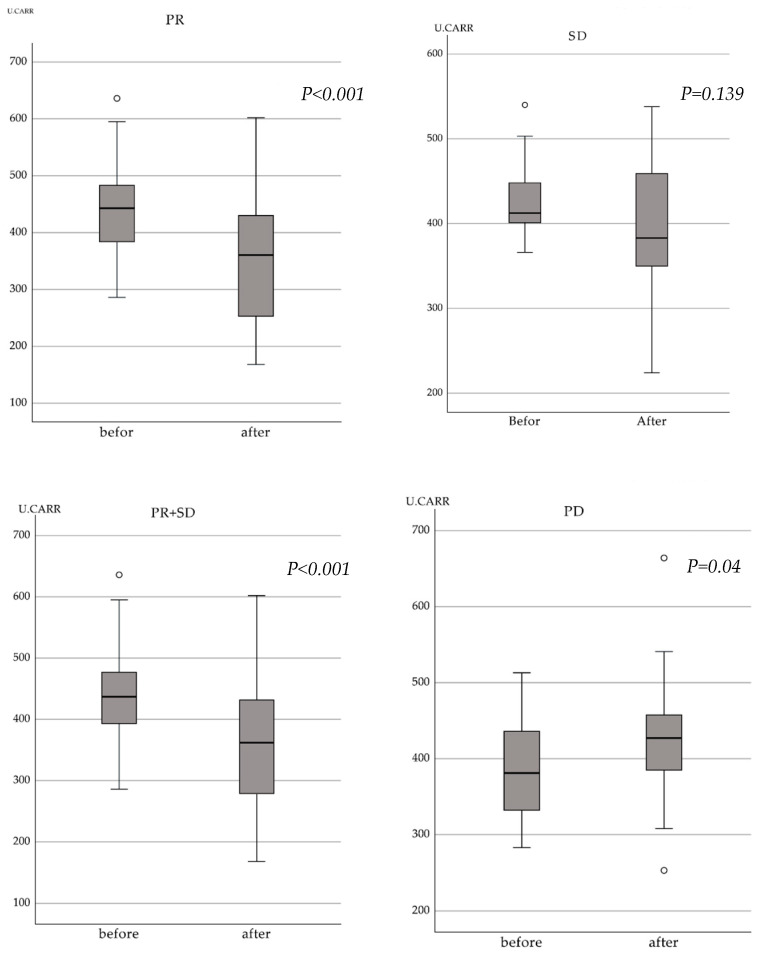
Change in d-ROM after 3 months. The d-ROM values of patients with PR decreased after treatment, while those of patients with PD increased. d-ROM—derivatives of reactive oxygen metabolites; PR—partial response; PD—progressive disease.

**Figure 2 cancers-15-05517-f002:**
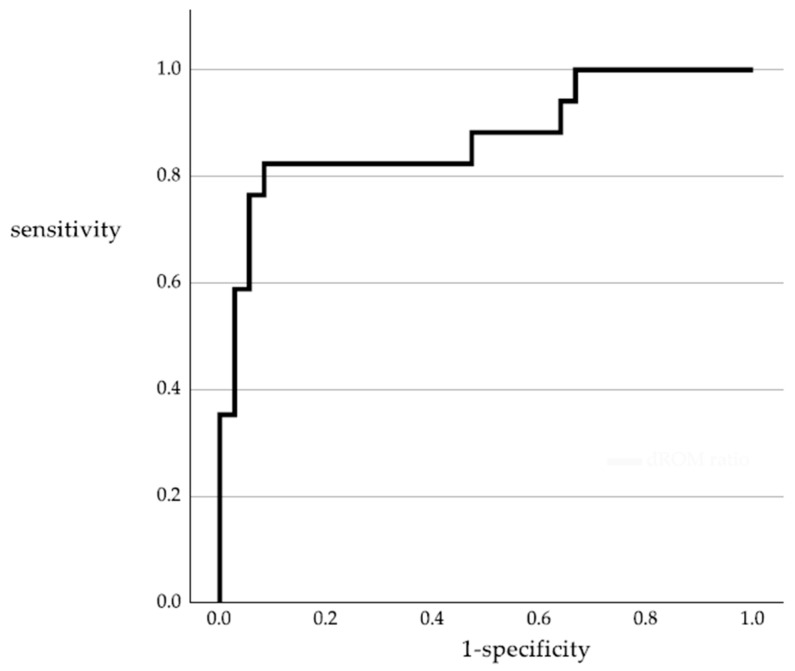
Receiver operating characteristic curve analysis of d-ROM ratio for predicting progression disease. Area under the curve = 0.874. d-ROM—derivatives of reactive oxygen metabolites.

**Table 1 cancers-15-05517-t001:** Demographic and clinical characteristics of patients with stage IV colorectal cancer (n = 53).

Characteristic	Number of Patients (%)
Sex (%)	
Male/female	30 (56.6)/23 (43.4)
Median age, years (range)	69 (38–93)
Location (%)	
Colon/rectum	30 (56.6)/23 (43.4)
Histological type (%)	
Well/moderate/poor	15 (28.3)/30 (56.6)/8 (15.1)
Primary tumor status (%)	
Resected/not resected	44 (83.0)/9 (17.0)
Metastatic presentation (%)	
Metachronous/synchronous	6 (11.3)/47 (88.7)
History of anticancer drug use (%)	
No/yes	37 (69.8)/16 (30.2)
Metastatic organs (%)	
Single/multiple	28 (52.8)/25 (47.2)

**Table 2 cancers-15-05517-t002:** Anticancer drug treatments.

Anticancer Drug	Number of Patients
Doublet oxaliplatin-based regimen (targeting agent used)	33 (23)
Doublet irinotecan-based regimen (targeting agent used)	5 (4)
Triplet regimen (targeting agent used)	4 (3)
5-Fluorouracil-based oral drug (targeting agent used)	5 (2)
Trifluridine/tipiracil hydrochloride (targeting agent)	2 (2)
No treatment	4

**Table 3 cancers-15-05517-t003:** Tumor RECIST evaluation after 3 months of treatment.

Evaluation by RECIST	Number of Patients (%)
Complete response	0 (0)
Partial response	26 (49.0)
Stable disease	10 (18.9)
Progressive disease	17 (32.1)

RECIST—Response Evaluation in Solid Tumors.

**Table 4 cancers-15-05517-t004:** d-ROM levels before and after 3 months of treatment.

Total	53 Cases
Levels at the start of treatment (median)	283–636 (424)
Levels after 3 months of treatment (median)	168–664 (375)
d-ROM ratio ≤ 1	36 cases
	(RECIST non-PD: 33 cases; PD: 3 cases)
d-ROM ratio > 1	17 cases
	(RECIST non-PD: 3 cases; PD: 14 cases)

d-ROM—derivatives of reactive oxygen metabolites; d-ROM ratio—post-treatment: pre-treatment d-ROM levels.

**Table 5 cancers-15-05517-t005:** Multivariate linear regression analysis of the d-ROM ratio in patients with metastatic colorectal cancer.

Independent Variables	Multivariate Linear Regression	
	β (95% CI)	*p*-Value
RECIST criteria (Non-PD vs. PD)	0.416 (0.279–0.555)	<0.001

d-ROM—derivatives of reactive oxygen metabolites; β—regression coefficient; CI—confidence interval; RECIST—Response Evaluation in Solid Tumors; PD—progressive disease.

**Table 6 cancers-15-05517-t006:** Accuracy of d-ROM ratio measurement for predicting disease progression.

	Sensitivity	Specificity	Positive Predictive Value	Negative Predictive Value	Diagnostic Accuracy
d-ROMs ratio > 1	82.4%	91.7%	82.4%	91.7%	88.7%

## Data Availability

All data included in this study are available upon request by contact with the corresponding author.

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
