# Peer review of "Monitoring Metastatic Colorectal Cancer Progression According to Reactive Oxygen Metabolite Derivative Levels"

_cancers, 2023, doi:10.3390/cancers15235517_

Round 1
Reviewer 1 Report
Comments and Suggestions for Authors
The present manuscript explains the role of d-ROM levels in monitoring metastatic colorectal cancer progression.
The work was introduced well and the objective of the present study was explained well.
The methodology adopted and the study population was found to be appropriate.
The data were analysed well and the authors followed the appropriate statistical methods for analyzing the data.
The results were presented with relevant tables and figures. If the authors provide better resolution images, it will be good for the readers.
The results were discussed well with relevant literature support.
The conclusion part may be improved.
The authors need to check the references and old references may be replaced with newer one.
Reference No.8 should be checked and modified as per the journal format.
Reviewer 2 Report
Comments and Suggestions for Authors
Dear authors,
The research article “Monitoring Metastatic Colorectal Cancer Progression According to Reactive Oxygen Metabolite Derivative Levels” by Sawai et al. is interesting, and I believe that readers would benefit from it. Having said that, your original submission has some potential for improvement, and some of the issues need to be addressed, for which I recommend minor revision. I've included a few of them below:
A brief description of the role of oxidative stress and cancer/colorectal cancer development could be included in the introduction section:
https://www.ncbi.nlm.nih.gov/pmc/articles/PMC8996905/
https://www.frontiersin.org/articles/10.3389/fendo.2023.1217165/full
https://www.ncbi.nlm.nih.gov/pmc/articles/PMC10135609/
https://www.ncbi.nlm.nih.gov/pmc/articles/PMC7698080/
https://www.mdpi.com/2072-6694/14/14/3525
https://ar.iiarjournals.org/content/37/9/4759
https://www.hindawi.com/journals/bmri/2013/725710/
Please provide better images (with higher resolution) for Figures 1 and 2.
I was wondering if the authors obtained any informed consent for this trial (please see the details https://www.ncbi.nlm.nih.gov/pmc/articles/PMC5980471/)
Reviewer 3 Report
Comments and Suggestions for Authors
This is a novel and interesting approach
My only concern is the machine itself being termed carpe diem
